# Missed Down Syndrome Cases after First Trimester False-Negative Screening—Lessons to be Learned

**DOI:** 10.3390/medicina56040199

**Published:** 2020-04-23

**Authors:** Anca Angela Simionescu, Ana Maria Alexandra Stanescu

**Affiliations:** 1Filantropia Hospital, Department of Obstetrics and Gynecology, University of Medicine and Pharmacy “Carol Davila”, 050474 Bucharest, Romania; 2Department of Family Medicine, University of Medicine and Pharmacy “Carol Davila”, 050474 Bucharest, Romania; alexandrazotta@yahoo.com

**Keywords:** prenatal diagnosis, Down syndrome, screening tests, software, biochemical markers

## Abstract

*Background and Objectives*: Here, we performed a descriptive analysis of Down syndrome (DS) cases that were misdiagnosed and/or false-negative diagnosed after first trimester traditional screening via risk evaluation using ultrasound, biochemical markers, and different software programs. Our objective was to demonstrate the clear need to improve the application of prenatal DS screening programs using standardized ultrasound measurements, accurate pregnancy dating, analytical immunoassay performance, and properly selected medians. *Materials and Methods*: We performed a database search for the period 2010–2015 to analyze DS cases that were false-negative diagnosed after the first trimester of pregnancy, before the introduction of cell free fetal DNA-based tests by Romanian laboratories in 2015. First-trimester screening was performed using two software programs for prenatal DS risk calculation: Astraia and Prisca. The rationale for using both software programs was to assess the full risk using the maternal age combined test (based on nuchal translucency thickness, nasal bone, ductus venosus flow, tricuspid flow, free beta-human chorionic gonadotropin level, and serum pregnancy-associated plasma protein-A) and, in some cases, the triple test. *Results*: We identified seven DS cases that exhibited low risk for trisomy 21, and 6540 cases with a low risk for trisomy 21 and euploid fetus in the first trimester. Using Astraia software, 14 cases were diagnosed, and three cases were missed after risk calculation. Using Prisca software, four cases were missed. Additionally, one neonate had a missed prenatal diagnosis of atrio-ventricular canal defect. *Conclusion*: In Romania, the evaluation of DS risk depends on patient choice (without knowing the accuracy of the utilized tests) and on the operators’ skills. Both Astraia and Prisca software were developed by experts, who can prove their performance in DS screening. However, even in an ideal situation, false-negative results are possible. The application of first and second-trimester combined screening based on biochemical markers could be improved by the implementation of standardized protocols, professional guidelines for test application, and audit control.

## 1. Introduction

Down syndrome (DS; trisomy 21) is caused by an additional chromosome 21, and is the most heavily studied autosomal trisomy. Half of DS cases are associated with cardiac or digestive malformations, and children born with DS have low intelligence and carry risks of leukemia and Alzheimer’s disease [1,2,3,4,5]. The worldwide DS incidence is estimated to be between one in 1000 and one in 1100 live births [6]. Despite the introduction of DS prenatal screening tests, the DS incidence at birth in Romania has remained fairly constant, e.g., one in 1863 births in 2015, one in 1975 births in 2018, and one in 1606 births in 2000 [7,8].

The accuracy of first-trimester DS screening tests has been demonstrated in large prospective international studies, with a combined test showing over 90% detection of T21, and a false-positive rate of 3–5% [9,10,11,12]. Additionally, several prospective and retrospective multicentric studies have shown the accuracy of advanced first-trimester screening using different software programs, and different test strategies. DS risk calculation has a sensitivity between 69.60% and 81.4% and a specificity of 95.14% when using nuchal translucency (NT) alone or combined with biochemical parameters, including levels of free beta-human chorionic gonadotropin (beta-hCG) and serum pregnancy-associated plasma protein-A (PAPP-A) [9,13,14,15,16]. The rate of invasive testing has been decreased by the introduction of cell-free fetal DNA-based tests (cffDNA) at 11–22 gestational weeks, as a primary screening method or after intermediate risk calculation [17]. DS detection rates are reportedly similar between non-invasive cffDNA-based testing and invasive testing based on cytogenetic analyses [18]. We would expect a significant improvement in DS screening as it becomes more widely available.

However, no data are available regarding the true value of national prenatal screening for DS, or how it has improved after the introduction of new screening tests. Specifically, we lack systematic data on cffDNA use, invasive test numbers, and results in hospitals or in the Romanian population.

DS screening programs are important because the family can be informed of a diagnosis, and therapeutic interruption of pregnancy can be performed [9] at under 24 weeks in accordance with the Penal Code of Romania. In Romania, the prenatal DS screening approach is based on a combination of maternal age, ultrasound parameters, and maternal biochemical serum markers during the first or second trimester of pregnancy, analyzed using different software programs, and with some data adjusted for maternal weight, ethnicity, smoking status, and patient history. However, there is no official national screening program, and the doctors or hospital staff are free to choose which biochemical marker and software to use for quantitative DS risk assessment. Since 2019, national guidelines have recommended the use of DS screening tests, and periodic evaluation of laboratories for measurement of PAPP-A and beta-hCG, but they do not specify which software should be used for risk calculation [19]. The biochemical serum markers used to screen for aneuploidies can be measured using different laboratory instruments, e.g., Kryptor (Brahms), Delphia Express (Perkin Elmer), Cobas (Roche), or Immulite (Siemens). Biochemical serum marker concentrations vary throughout gestation, and are thus converted to gestational age-adjusted multiples of the median (MoM) values with the selected software. A small percentage of ultrasound examiners (less than 0, 5% percent of sonographers, 33 obstetricians in 2019) have been trained, certified, and accredited by the Fetal Medicine Foundation (FMF), London, for measurements of NT, nasal bone, tricuspid flow, and ductus venosus flow (DV) between 11+0 to 13+6 gestational weeks (www.fetalmedicine.org). However, there is no national accreditation or quality control evaluation. Moreover, there is no external software quality assurance program.

In the first trimester of pregnancy, individual risk assessment of fetal Down syndrome can be performed based on maternal age and ultrasound parameters (NT thickness, nasal bone, DV, and tricuspid flow), using Astraia software, and following Fetal Medicine Foundation protocols. Risk assessment based on biochemical parameters, including levels of free beta-hCG and serum PAPP-A at between 11+0 and 13+6 weeks of gestation is adjusted for maternal characteristics, gestational age and patient history. When using Astraia software, maternal serum levels of free beta-hCG and PAPP-A are measured using the Kryptor rapid random access immunoassay analyser (Brahms Diagnostica AG, Berlin) or the Delphia Express system (Perkin Elmer, USA) using time-resolved amplified cryptate emission technology. Additionally, before 2015, a first-trimester combined test using Prisca software was used to determine risk based on maternal age, gestational age, nuchal translucency, and first trimester biochemical markers (hCG and PAPP-A) between 11+0 and 13+6 weeks of gestation or second trimester biochemical markers. When using Prisca software, NT is assessed between 10+0 and 13+6 weeks, and transformed into MoM.

For patients who do not undergo screening during the first trimester, second trimester risk assessment (the “triple test”) is based on maternal age combined with biochemical parameters, including alpha-fetoprotein (AFP), beta-hCG, and unconjugated estriol (uE3) between 15 and 18 weeks of gestation. Alternatively, second-trimester risk assessment may include maternal age combined with both biochemical parameters and ultrasound parameters for gestational age, e.g., biparietal diameter (BPD).

The results of first trimester screenings include an initial (background) risk based on maternal age at the time of analysis, and an individual risk at the time of analysis calculated by a sonography specialist or at a laboratory using Astraia software, which is based on maternal age-adjusted parameters, patient history, biochemical markers, and ultrasound parameters (including fetal heart rate) in the first trimester of pregnancy. The results of the DS screening using Prisca software are based on age-related risk, measurement of NT (and since 2017, CRL, nasal bone and first trimester PAPP-A and beta-hCG), biochemical risk in the second trimester (beta-hCG, AFP, and uE3), or combined risk based on BPD, beta-hCG AFP, and uE3 (in the second trimester). Risks are calculated by a medical technologist from the laboratory department. In recent years, some doctors have recommended screening for Down syndrome using non-invasive prenatal testing (NIPT) via massive parallel sequencing for cffDNA, instead of first or second trimester individual risk assessment. Prior to the availability of NIPT, risk was classified as positive or negative at a level depending on the software. Since NIPT introduction, the calculated risk has been classified as positive, negative, or borderline. For patients with negative risk, second-trimester ultrasound is recommended. Patients with positive risk are advised to undergo fetal karyotyping. For the majority of patient with intermediate risk, NIPT is recommended. Thus, overall, DS screening depends on the doctor’s knowledge, the patient’s access to laboratory and ultrasound parameter measurements, biochemical marker analysis, and software use and selection. All screening software have been developed by experts, who have proven their performance for DS screening. 

In the present study, we retrospectively reviewed first-trimester and mid-trimester DS risks, and performed a descriptive analysis of, and software calculations for, seven cases with false-negative first-trimester DS screening test results from 2010–2015 (before the introduction of cffDNA tests) at the Filantropia Hospital, in Bucharest. Our principal objective was to demonstrate the need to improve the trisomy 21 screening program based on the standardized use of ultrasound measurements, biochemical markers, and software selection. A secondary objective was to demonstrate the usefulness of large primary care databases for pregnancy care and infant data. 

## 2. Materials and Methods

We performed a database search in the Births and Ultrasounds Hospital Registry, and in hospital databases, for the period 2010–2019. We identified cases of Down syndrome that were false negative diagnosed after the first or second trimester of pregnancy, among both abortions and live births in our hospital. At Filantropia Hospital in Bucharest, we identified 39,032 live births, 10,102 first-trimester ultrasounds with DS risk assessment during the first trimester of pregnancy, and 14,600 s trimester screening ultrasounds between 2010–2019. The cases between 2010 and 2015 (before cffDNA testing was introduced) included 20,501 live births, 5114 first-trimester ultrasounds with DS risk assessment during the first trimester of pregnancy, and 7600 second-trimester screening ultrasounds.

In our hospital, ultrasound measurements during the first trimester of pregnancy were performed by certified (and sometimes accredited) sonographers according to FMF standards. The same doctors performed second-trimester ultrasound examinations. We conducted traditional first-trimester screening, using maternal age at the time of measurement, of standardized ultrasound parameters (NT, nasal bone, DV, tricuspid flow, and fetal heart rate) and MoM values for biochemical markers (free beta-hCG level and serum PAPP-A), using ASTRAIA software. Biochemical markers were assessed in an FMF-certified laboratory. Before 2015, a high positive risk was defined as a calculated risk greater than 1 in 100, and a low negative risk as a calculated risk below 1 in 100. After 2015 (when cffDNA tests were introduced), a high positive risk was defined as a calculated risk greater than 1 in 100, a borderline risk as between 1 in 101–2500, and a low negative risk was defined as a calculated risk greater than 1 in 2500. In cases of positive risk, we recommended karyotyping, chorionic villus sampling between 11–14 weeks, or amniocentesis between 16–18 weeks. In cases where the MoM for PAPP-A was <0.3, or structural abnormalities were found, we recommended amniocentesis to enable classic karyotyping and CGH array. At our hospital, patients received an explanation of what screening tests meant, and of the fact that DS is diagnosed by fetal karyotype. Notably, some patients enrolled in this study (who gave birth at our hospital) had undergone ultrasounds and risk calculation during the first and second trimesters at other medical centers, as well as risk calculations based on bi-test or triple test, most often also using Astraia for laboratory use or the PRISCA method. All of the patients examined outside of our center were categorized as low risk or high risk for DS after the result was received. Some of these patients only gave birth in our hospital, while others underwent second-trimester screening in our hospital. In the present study, we retrospectively reviewed first trimester and mid-trimester DS risks, and performed a descriptive analysis of seven cases with false-negative first trimester DS screening test results. The costs of biochemical marker analysis were paid by the family or local laboratory programs. The cost of cffDNA testing ($400–1100) were paid by the family. National Insurance contributions covered the first-trimester ultrasound. 

## 3. Results

At our hospital between 2010–2015, there were 20,501 recorded live births, and 21 cases with DS, including five newborns and 16 cases with termination of pregnancy (TOP). During this period, 5114 first-trimester ultrasound scans and 7600 s-trimester ultrasound screenings were performed. The screening results indicated low risk of trisomy 21 in seven cases of Down syndrome, as well as 6540 cases of low-risk trisomy 21 and euploid fetus in the first trimester. Using Astraia computer software and second-trimester ultrasound, we correctly diagnosed 16 cases of DS. Fourteen cases had **a** positive risk after the first-trimester combined risk assessment (false positive). In two cases (numbers 1 and 6), Prisca software calculated negative DS screening results in the first trimester.

Table 1 presents the eight cases with DS and false-positive risk calculation after first-trimester ultrasound risk assessment.

In two cases (numbers 2 and 3), DS was diagnosed during the second trimester of pregnancy, and was followed by TOP. In case number 2 (a 33-year-old primigravida), amniocentesis was performed due to abnormal ductus venosus and patient worry. In case number 3 (a 39-year-old primigravida), amniocentesis was performed due to cerebral ventriculomegaly, corpus callosum agenesis, and polyhydramnios at 21 weeks of gestation. In case number 1, a 26-year-old primigravida was monitored after second-trimester ultrasound due to nasal bone of 4.5 mm, and the fetal stomach showing reduced filling (normal amniotic fluid, and suspicion of esophageal atresia that was not confirmed postnatally). The patient declined the proposed amniocentesis, and the baby was born with DS and without structural abnormalities. In case number 4, a 39-year-old primigravida was transferred from a country hospital due to intrauterine growth restriction (with altered Doppler parameters, and hydrops fetalis) to our level III maternity department to give birth. 

Case number 5 (a 42-year-old primigravida) exhibited NT of 1.5 mm at 11 weeks and five days and underwent first-trimester DS risk assessment using Prisca software. In cases 5, 6, and 8, second-trimester ultrasound revealed no structural abnormalities. Unfortunately, in case number 6, an atrioventricular canal hearth defect was missed.

In case number 7, first-trimester screening was performed in accordance with FMF guidelines at another medical center. Although the screening results were negative, the NT and ductus venosus flow were abnormal. No structural abnormalities were observed in the second trimester or at birth. In cases 1, 3, 5, 6, and 7, the second-trimester ultrasound was performed at our center.

## 4. Discussion

Our present findings raise questions about whether patients are likely to get correct DS risk assessment. European guidelines for clinical prenatal diagnosis practice recommend that health professionals may provide prenatal diagnosis counselling, presenting additional challenges for ensuring good clinical practice and equity of care [20]. Large prospective studies have shown improved DS detection rates and significantly reduced false-positive rates [21,22,23].

To our knowledge, this was the first investigation of the clinical application of first-trimester and second-trimester DS screening software using ultrasound parameters and biochemical markers in Romania, excluding the use of cffDNA testing, which is very expensive. In Romania, the monthly salary is $500–600, and DS screening is expensive ($50 for biochemical marker analysis, $150 for ultrasound, $400–1200 for cffDNA testing) and most often paid for by the patient. Moreover, the financial costs of caring for DS babies are paid by the family.

Obviously, screening tests cannot serve as a substitute for diagnostic tests, but the present study was focused on determining whether the application of DS screening tests could be improved. A few reports of DS screening performance have indicated good national results, although from a single medical unit. For example, one single-center audit in Romania has examined performance using different software programs, and reported similar results compared with international specialized centers [24]. Additionally, an analysis of the systematic application of FMF software in Filantropia Hospital between 2010–2019 revealed a prenatal aneuploidy detection rate of 87.5%, with a 3% false-positive rate, when using nuchal translucency and first-trimester biochemical markers. Notably, national studies have demonstrated that the majority of pregnancies identified by the combined test as being high risk for trisomy 21, 18, or 13 are euploid [25]. In a three-year study at a private Romanian hospital, DS screening of 6097 pregnant women from 2012–2015 revealed that 7% of pregnancies showed high risk for aneuploidies. A total of 408 amniocentesis were performed, including 131 based on bi-test screening, 258 on triple-test screening, 15 with maternal age under 35 years, and four due to personal history of aneuploidy, resulting in the diagnosis of 10 cases of DS and four cases of trisomy 18 [26]. 

All patients in our present descriptive study underwent first-trimester DS risk assessment, and second-trimester screening ultrasound. Of the seven misdiagnosed DS cases, cases 2 and 3 were correctly diagnosed during the second trimester. In case 2, although ultrasound screening indicated low risk, CVS was proposed because of NT at the 85th percentile and abnormal DV. Ductus venosus flow velocity waveform evaluation for DS screening test has almost the same detection rate as NT alone (75–76%), and evaluating both markers together increases the detection rate to 85%. When using NT, PAPP-A, and free beta-hCG, the addition of DV will increase the detection rate from 88% to 92% [27,28]. In case 3, amniocentesis was performed due to structural defects, demonstrating that ultrasound performed by a certified doctor is important for identifying borderline ventriculomegaly, corpus callosum agenesis, and other indications for fetal karyotyping. The ultrasound performer could have improved the accuracy of the calculated risk for a 39-year-old patient, based on DV and tricuspid flow, to increase the sensitivity of first-trimester screening. An abnormal Doppler DV waveform associated with advanced maternal age is a predictor of adverse outcome even with normal NT [29]. For the three cases evaluated with Prisca software (cases 1, 5, and 6), NT was measured at between 11+6 weeks and 12+1 weeks. For case 6, the missed atrioventricular canal defect could have prompted a recommendation of fetal karyotyping if noticed. For cases 1 and 5, risk calculation using Astraia software would also have predicted a low risk for DS.

The difference between Prisca and Astraia software is that Astraia includes adjustment of the ultrasound and biochemical parameters according to personal history and demographic and clinical characteristics. Additionally, Prisca software uses the MoM for nuchal translucency. Since 2003, in large prospective studies, Nicolaides and Spencer have demonstrated that the MoM approach for NT generates an overestimation of trisomy risk at 11 gestational weeks, and a considerable underestimation of risk at 13 gestational weeks. In contrast, the Delta NT approach provides accurate patient-specific DS risks. Moreover, the NT MoM approach for DS risk assessment can be considered inappropriate for several reasons: the distributions of NT MoM and log10 (NT MoM) were not Gaussian within the unaffected population, the standard deviations did not remain constant with gestation, and the median MoM among trisomy 21 pregnancies was not a constant proportion of the median among unaffected pregnancies [30]. In case number 7, findings of a 3.3-mm NT and abnormal DV led to recommendation of fetal karyotyping despite the “low risk “calculation. The recalculated DS risk using Astraia software was high (1/4).

Case number 4 could have been prenatally diagnosed by amniocentesis; however, diagnosis before birth was missed due to fetal degradation caused by intrauterine growth restriction with associated Doppler alterations.

Based on our findings, we are not advocating for the use of Astraia or Prisca software. The false-negative diagnosed cases are in accordance with the previously reported rates of false-negative cases in screening [31,32]. Rather, we found that this false negative case did not reflect a correct use of the screening.

Since the introduction of biochemical markers and software for DS risk calculation, they have been used in many centers and by most operators. Some doctors and medical laboratory centers are certified and accredited by the FMF for measuring ultrasound parameters and for DS risk calculation. However, not all measurements are performed by certified doctors, and the DS risk assessment is performed by the laboratory software. There are differences in accuracy when the risk is calculated immediately after ultrasound by a certified doctor compared to a later risk calculation by laboratory staff. There is currently no national audit or feedback regarding biochemical parameters and routine first-trimester nuchal translucency ultrasound images, ductus venosus, nasal bone, or tricuspid flow. Moreover, there are no complete databases of pregnancy care and infant data.

Nowadays, prenatal aneuploidies screening and diagnosis are routinely offered in antenatal care, and are considered important for managing risk pregnancies and allowing women to make informed choices about the termination of pregnancies affected by DS. A range of policies for prenatal diagnosis have been developed in different countries worldwide, including Romania. Since the introduction of screening policies, most European countries have reported a significant reduction in the rate of liveborns infants with Down syndrome [33,34,35].

In Romania, noninvasive measurements of ultrasound parameters combined with maternal serum markers during the first or second trimester of pregnancy have allowed extension of screening to mothers of all ages, with the use of different software programs for aneuploidy risk calculation. The biochemical markers used in the first trimester of pregnancy (8–13 weeks) are PAPP-A and beta-hCG, and those used in the second trimester of pregnancy (15–18 weeks) are beta-hCG, AFP, and uE3. Different software programs use different techniques for risk calculation using the requested biochemical markers, and present the results in MoM. There are technical standards and recommendations for prenatal DS screening, which include first-trimester biochemistry and/or ultrasound measurements for the first and second trimesters, and can help healthcare professionals and laboratories determine which techniques and screening protocols to choose and how to interpret the results [36]. It is expected that the integration of cffDNA data in the screening algorithm will increase the accuracy of aneuploidy detection, once the cffDNA testing costs becomes affordable or paid by National Insurance. However, even then, more accurate risk assessment will be needed to ensure judicious use of this expensive test. The main limitation of the present study was the small number of cases analyzed. The strengths of this study include the discussion of screening analysis using multiple software programs in a general population of pregnant women, and a comparison with the reported cases in other centers.

## 5. Conclusions

In conclusion, the stable number of newborns with DS in Romania over the last 20 years justifies the need for appropriate training for all doctors who perform screening ultrasounds, the use of biochemical markers in screening, as well as an audit of DS screening in the general population. Most commercial software packages rely on the same algorithm (overlapping multivariate Gaussian distributions) for risk calculation. Therefore, the quality of the results obtained depends on quality assurance, standardization of ultrasound measurements, accurate pregnancy dating, and analytical immunoassay performance. Screening quality might be improved by the use of several ultrasound parameters, proper median selection for MoM calculation, and the use of valid factors for MoM adjustment, which is widely accepted internationally.

## Figures and Tables

**Table 1 medicina-56-00199-t001:** The seven cases with DS and false-positive risk calculation after first-trimester ultrasound risk assessment.

Case Number	Maternal Age at Time of Risk Calculation (Years)	First Trimester Risk Calculation					Second Trimester Risk Calculation: Ultrasound Markers, Amniocentesis	Pregnancy Issue
		Low Risk Calculation, Software	NT (mm)/CRL (mm)	Abnormal DV, Tricuspid Flow, Other	Biochemical Parameters, MoM			
					HCG	PAPPA		
1	26	1/141, Prisca	1.0/55	U	3.92	1.17	Nasal bone = 4.5 mm; reduced stomach filling	Infant with T21
2	33	1/128, Astraia	2.2/60.1	Abnormal DV	1..535	0.55	Amniocentesis	TOP
3	39	1/232, Astraia	2.1/69.5	U	1.184	0.667	Corpus callosum agenesis; borderline ventriculomegaly; polyhydramnios; Amniocentesis	TOP
4	37	1/230, Astraia	1.2/70	U	U	U	Intracardiac echogenic focus; intrauterine growth restriction; abnormal Doppler; hydrops fetalis	Extremely premature birth T21
5	42	1/736, Prisca	1.5/52		0.739	0.694	Normal	Infant with T21
6	26	1/452, Prisca	U	U	U		Normal	Infant with T21 and atrioventricular canal defect
7	29	1/2744, Astraia	3.3/74 mm	Abnormal DV	U	U	Normal	Infant with T21

U, unknown; NT, nuchal translucency; CRL, crown-rump length; MoM, multiples of the median; HCG, free beta-human chorionic gonadotropin; PAPPA, pregnancy-associated plasma protein-A; T21, trisomy 21; DV, ductus venosus flow; TOP, termination of pregnancy.

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
