# Peer review of "Missed Down Syndrome Cases after First Trimester False-Negative Screening—Lessons to be Learned"

_medicina, 2020, doi:10.3390/medicina56040199_

Round 1

Reviewer 1 Report

This manuscript is well written and it is of scientific importance in order to improve DS screening in certain countries. The methods and results are very well described and presented. The discussion could be a slightly improved with a more in depth analysis of how the implementation of new screening quality standards suggested by the authors would have affected the 7 cases described as DS and false-positive risk calculations in the results section. I also recommend minor revisions to the language in order to correct some errors (line 183, etc...)

Thanks

Author Response

We wish to thank you for your insightful comments which we feel have substantially improved our manuscript . We appreciate the time and detail provided by you and I have incorporated the suggested changes into the manuscript to the best of my ability .

I have responded specifically to each suggestion below. To make the changes easier to identify where necessary , I have numbered them and I have put them in red into the manuscript.

Reviewer  1  :

(1) English language and style are fine/minor spell check required

The manuscript was checked by a professional English editing service. 

(2) The discussion could be a slightly improved with a more in depth analysis of how the implementation of new screening quality standards suggested by the authors would have affected the 7 cases described as DS and false-positive risk calculations in the results section.

The discussion was rewritten with more in depth analysis . I highlighted in the discussion how the correct application of DS screening tests and software could have improve the results.

Screening tests cannot serve as a substitute for diagnostic tests, but the present study was focused on determining whether the application of DS screening tests could be improved. A few reports of DS screening performance have indicated good national results, although from a single medical unit. For example, one single-center audit in Romania has examined performance using different software programs, and reported similar results compared with international specialized centers. Additionally, an analysis of the systematic application of FMF software in Filantropia Hospital between 2010–2019 revealed a prenatal aneuploidy detection rate of 87.5%, with a 3% false-positive rate, when using nuchal translucency and first-trimester biochemical markers. Notably, national studies have demonstrated that the majority of pregnancies identified by the combined test as being high risk for trisomy 21, 18, or 13 are euploid. In a 3-year study at a private Romanian hospital, DS screening of 6097 pregnant women from 2012–2015 revealed that 7% of pregnancies showed high risk for aneuploidies. A total of 408 amniocentesis were performed—including 131 based on bi-test screening, 258 on triple-test screening, 15 with maternal age under 35 years, and 4 due to personal history of aneuploidy—resulting in the diagnosis of 10 cases of DS and 4 cases of trisomy 18.

All patients in our present descriptive study underwent first-trimester DS risk assessment, and second-trimester screening ultrasound. Of the seven misdiagnosed DS cases, cases 2 and 3 were correctly diagnosed during the second trimester. In case number 2 , although ultrasound screening indicated low risk, CVS was proposed because of NT at the 85th percentile and abnormal DV. Ductus venosus flow velocity waveform evaluation for DS screening test has almost the same detection rate as NT alone (75–76%), and evaluating both markers together increases the detection rate to 85%. When using NT, PAPP-A, and free beta-hCG, the addition of DV will increase the detection rate from 88% to 92%. In case 3, amniocentesis was performed due to structural defects, demonstrating that ultrasound performed by a certified doctor is important for identifying borderline ventriculomegaly, corpus callosum agenesis, and other indications for fetal karyotyping. The ultrasound performer could have improved the accuracy of the calculated risk for a 39-year-old patient, based on DV and tricuspid flow, to increase the sensitivity of first-trimester screening. An abnormal Doppler DV waveform associated with advanced maternal age is a predictor of adverse outcome even with normal. For the three cases evaluated with Prisca software (cases 1, 5, and 6), NT was measured at between 11+6 weeks and 12+1 weeks. For case 6, the missed atrioventricular canal defect could have prompted a recommendation of fetal karyotyping if noticed. For cases 1 and 5, risk calculation using Astraia software would also have predicted a low risk for DS.

The  difference between Prisca and Astraia software is that Astraia includes adjustment of the ultrasound and biochemical parameters according to personal history and demographic and clinical characteristics. Additionally, Prisca software uses the MoM for nuchal translucency. Since 2003, in large prospective studies, Nicolaides and Spencer have demonstrated that the MoM approach for NT generates an overestimation of trisomy risk at 11 gestational weeks, and a considerable underestimation of risk at 13 gestational weeks. In contrast, the Delta NT approach provides accurate patient-specific DS risks. Moreover, the NT MoM approach for DS risk assessment can be considered inappropriate for several reasons: the distributions of NT MoM and log10(NT MoM) were not Gaussian within the unaffected population, the standard deviations did not remain constant with gestation, and the median MoM among trisomy 21 pregnancies was not a constant proportion of the median among unaffected pregnancies . In case number 7, findings of a 3.3-mm NT and abnormal DV led to recommendation of fetal karyotyping despite the “ low risk “calculation. The recalculated DS risk using Astraia software was high (1/4).

Case number 4 could have been prenatally diagnosed by amniocentesis; however, diagnosis before birth was missed due to fetal degradation caused by intrauterine growth restriction with associated Doppler alterations.

Based on our findings, we are not advocating for the use of Astraia or Prisca software. The false-negative diagnosed cases are in accordance with the previously reported rates of false-negative cases in screening[31,32] . Rather, we found that this false negative cases did not reflect a correct use of the screening. 

Nowadays, prenatal aneuploidies screening and diagnosis are routinely offered in antenatal care, and are considered important for managing risk pregnancies and allowing women to make informed choices about the termination of pregnancies affected by DS . A range of policies for prenatal diagnosis have been developed in different countries worldwide, including Romania. Since the introduction of screening policies , most European countries have reported a significant reduction in the rate of liveborns infants with Down syndrome.

In Romania, noninvasive measurements of ultrasound parameters combined with maternal serum markers during the first or second trimester of pregnancy have allowed extension of screening to mothers of all ages, with the use of different software programs for aneuploidy risk calculation. The biochemical markers used in the first trimester of pregnancy (8–13 weeks) are PAPP-A and beta-hCG, and those used in the second trimester of pregnancy (15–18 weeks) are beta-hCG, AFP, and uE3. Different software programs use different techniques for risk calculation using the requested biochemical markers, and present the results in MoM. There are technical standards and recommendations for prenatal DS screening, which include first-trimester biochemistry and/or ultrasound measurements for the first and second trimesters, and can help healthcare professionals and laboratories determine which techniques and screening protocols to choose and how to interpret the results . It is expected that the integration of cffDNA data in the screening algorithm will increase the accuracy of aneuploidy detection, once the cffDNA testing costs becomes affordable or paid by National Insurance. However, even then, more accurate risk assessment will be needed to ensure judicious use of this expensive test. The main limitation of the present study was the small number of cases analyzed. The strengths of this study include the discussion of screening analysis using multiple software programs in a general population of pregnant women, and comparison with the reported cases in other centers.

Reviewer 2 Report

The present is an interesting study on missed DS cases after non-invasive first trimester screening in Romania. The authors describe 7 cases misdiagnosed between 2010-2015 at the Filantropia Hospital in Bucharest. Overall, the manuscript requires extensive English editing to avoid replication of sentences and to be more focused to the point. In addition, the authors must be able to put their research within the global context of prenatal non-invasive screening.

Major points:

1) A Table detailing differences and similarities between ASTRAIA and PRISCA softwares is required.

2) The authors must clarify the reason to restrict their search to the period 2010-2015. We are now in 2020, and also data from 2015 to 2020 would be of interest. Particularly, are PRISCA and ASTRAIA still used in your hospitals? Which one works better? Is there any difference in DS mis-diagnosis after the recent introduction of NIPT technologies? 

3) Please discuss your data in a broader context. What is the rate of DS misdiagnosis in Romania (not only in your Hospital, but generally). You found 7 misdiagnosed cases out of several screenings. It is however known that the non-invasive prenatal testing can lead to either false-positive or false-negative results. Is the frequency of false-negative detected at your Hospital greater or smaller than those observed outside Romania, i.e. the general frequencies reported in the literature? 

4) In general, you must change the prospective of your article and be able to describe your data in a more general context, making comparisons with other hospitals in your Country as well as outside your Country. Is the rate of mis-diagnoses in your Country really higher than in other Countries? Furthermore, you must show data and discuss if the technological improvements (NIPT techniques) and the Medical training have changed the situation in the years 2015-2020, both in your hospital as well as in general.

Author Response

Reviewer 2 comments

Thank you for your time, effort and expert evaluation.

(1) Extensive editing of English language and style required

The original manuscript and this revised manuscript  were checked by a professional English editing service. 

(2) The authors must be able to put their research within the global context of prenatal non-invasive screening and (3) The authors must clarify the reason to restrict their search to the period 2010-2015. We are now in 2020, and also data from 2015 to 2020 would be of interest. Particularly, are PRISCA and ASTRAIA still used in your hospitals? Which one works better? Is there any difference in DS mis-diagnosis after the recent introduction of NIPT technologies?

We have revised the data between 2010-2019 and we excluded the period after 2015 , after the introduction of cff DNA because  our objective was to demonstrate the clear need to improve  the application of prenatal DS screening programs using standardized ultrasound measurements, accurate pregnancy dating, analytical immunoassay performance, and properly selected medians. We lake data about cffDNA in our registries . Also we lack systematic data on invasive test numbers, and results in hospitals or in the Romanian population (line 60-61).We use Astraia and FMF guidelines in Filantropia hospital but Prisca software is used by laboratory .

I am not able to demonstrate the difference in the studied population when cffDNA was performed. But, all 7 cases were classified low risk after first trimester DS screening and two cases with structural abnormalities which indicate fetal karyotype.

(4) A Table detailing differences and similarities between ASTRAIA and PRISCA softwares is required.

Comparing differences and similarities for DS risk calculation software packages is difficult for a  number  of  reasons.  Both  the  Astraia  software and Prisca software  are  developed  by  experts,  who  are  able  to  analyze  and  reanalyze  large  data-sets. The  commercial  derivates  of  these  software  packages,  that  are  tailor-made  for  use  in  a  routine screening setting, are not. Thus, for parties using one software  package,  it  is  extremely  difficult  to  reanalyze  their  data  using  another.  A    “  combined  risk  ”  is  the  product  of  a  prior  risk,  the  LR  of  biochemical  parameters  and  the  LR  of  the  NT  measurement. Since after harmonization prior risks are similar and the biochemical  LR  is  always  calculated  with  the  Prisca  software,  the remaining discrepancies can be reduced to the LR of the NT measurement. The Prisca and Astraia software use the same CRL-NT  reference  curve  but  different  methods  to  convert  a  NT into a LR. In the Prisca software this is done by calculating the NT-MoM, while in Astraia the Δ-NT is used. A large comment about NT -MoM is in line  . The  screening  performance  indicators  of  both  software  packages is  quite  similar.

I insert in text the differences and similarities when use the two software : line 82-90; 90-94; 101-109; 244-256; 260.

(5) Please discuss your data in a broader context. What is the rate of DS misdiagnosis in Romania (not only in your Hospital, but generally). You found 7 misdiagnosed cases out of several screenings. It is however known that the non-invasive prenatal testing can lead to either false-positive or false-negative results. Is the frequency of false-negative detected at your Hospital greater or smaller than those observed outside Romania, i.e. the general frequencies reported in the literature?

The DS incidence at birth in Romania has remained fairly constant—for example, 1/1863 births in 2015, 1/1975 birth in 2018 , and 1/1606 births in 2000.

An analysis of the systematic application of FMF software -Astraia software in Filantropia Hospital between 2010–2019 revealed a prenatal aneuploidy detection rate of 87.5%, with a 3% false-positive rate, when using nuchal translucency and first-trimester biochemical markers.

A few reports of DS screening national performance have indicated good national results, although from a single medical unit. For example, one single-center audit in Romania has examined performance using different software programs, and reported similar results compared with international specialized centers (Nemescu et al). In an important Romanian hospital Radoi et al.   have demonstrated that the majority of pregnancies identified by the combined test as being high risk for trisomy 21, 18, or 13 are euploid. And in one important private hospital in Bucharest, Romania , in a 3-year of DS screening of 6097 pregnant women from 2012–2015 revealed that 7% of pregnancies showed high risk for aneuploidies. A total of 408 amniocentesis were performed—including 131 based on bi-test screening, 258 on triple-test screening, 15 with maternal age under 35 years, and 4 due to personal history of aneuploidy—resulting in the diagnosis of 10 cases of DS and 4 cases of trisomy 18!

(6) In general, you must change the prospective of your article and be able to describe your data in a more general context, making comparisons with other hospitals in your Country as well as outside your Country. Is the rate of mis-diagnoses in your Country really higher than in other Countries? Furthermore, you must show data and discuss if the technological improvements (NIPT techniques) and the Medical training have changed the situation in the years 2015-2020, both in your hospital as well as in general.

The discussion was rewritten , with a more in depth analysis of how the correct use of screening quality standards suggested would have affected the 7 cases described as DS and false-positive risk calculations in the results section. Unfortunately , Medical training was not associated with the decrease of number of DS newborns .

Screening tests cannot serve as a substitute for diagnostic tests, but the present study was focused on determining whether the application of DS screening tests could be improved. A few reports of DS screening performance have indicated good national results, although from a single medical unit. For example, one single-center audit in Romania has examined performance using different software programs, and reported similar results compared with international specialized centers. Additionally, an analysis of the systematic application of FMF software in Filantropia Hospital between 2010–2019 revealed a prenatal aneuploidy detection rate of 87.5%, with a 3% false-positive rate, when using nuchal translucency and first-trimester biochemical markers. Notably, national studies have demonstrated that the majority of pregnancies identified by the combined test as being high risk for trisomy 21, 18, or 13 are euploid. In a 3-year study at a private Romanian hospital, DS screening of 6097 pregnant women from 2012–2015 revealed that 7% of pregnancies showed high risk for aneuploidies. A total of 408 amniocentesis were performed—including 131 based on bi-test screening, 258 on triple-test screening, 15 with maternal age under 35 years, and 4 due to personal history of aneuploidy—resulting in the diagnosis of 10 cases of DS and 4 cases of trisomy 18.

All patients in our present descriptive study underwent first-trimester DS risk assessment, and second-trimester screening ultrasound. Of the seven misdiagnosed DS cases, cases 2 and 3 were correctly diagnosed during the second trimester. In case number 2 , although ultrasound screening indicated low risk, CVS was proposed because of NT at the 85th percentile and abnormal DV. Ductus venosus flow velocity waveform evaluation for DS screening test has almost the same detection rate as NT alone (75–76%), and evaluating both markers together increases the detection rate to 85%. When using NT, PAPP-A, and free beta-hCG, the addition of DV will increase the detection rate from 88% to 92%. In case 3, amniocentesis was performed due to structural defects, demonstrating that ultrasound performed by a certified doctor is important for identifying borderline ventriculomegaly, corpus callosum agenesis, and other indications for fetal karyotyping. The ultrasound performer could have improved the accuracy of the calculated risk for a 39-year-old patient, based on DV and tricuspid flow, to increase the sensitivity of first-trimester screening. An abnormal Doppler DV waveform associated with advanced maternal age is a predictor of adverse outcome even with normal. For the three cases evaluated with Prisca software (cases 1, 5, and 6), NT was measured at between 11+6 weeks and 12+1 weeks. For case 6, the missed atrioventricular canal defect could have prompted a recommendation of fetal karyotyping if noticed. For cases 1 and 5, risk calculation using Astraia software would also have predicted a low risk for DS.

The  difference between Prisca and Astraia software is that Astraia includes adjustment of the ultrasound and biochemical parameters according to personal history and demographic and clinical characteristics. Additionally, Prisca software uses the MoM for nuchal translucency. Since 2003, in large prospective studies, Nicolaides and Spencer have demonstrated that the MoM approach for NT generates an overestimation of trisomy risk at 11 gestational weeks, and a considerable underestimation of risk at 13 gestational weeks. In contrast, the Delta NT approach provides accurate patient-specific DS risks. Moreover, the NT MoM approach for DS risk assessment can be considered inappropriate for several reasons: the distributions of NT MoM and log10(NT MoM) were not Gaussian within the unaffected population, the standard deviations did not remain constant with gestation, and the median MoM among trisomy 21 pregnancies was not a constant proportion of the median among unaffected pregnancies . In case number 7, findings of a 3.3-mm NT and abnormal DV led to recommendation of fetal karyotyping despite the “ low risk “calculation. The recalculated DS risk using Astraia software was high (1/4).

Case number 4 could have been prenatally diagnosed by amniocentesis; however, diagnosis before birth was missed due to fetal degradation caused by intrauterine growth restriction with associated Doppler alterations.

Based on our findings, we are not advocating for the use of Astraia or Prisca software. The false-negative diagnosed cases are in accordance with the previously reported rates of false-negative cases in screening[31,32] . Rather, we found that this false negative cases did not reflect a correct use of the screening. 

Nowadays, prenatal aneuploidies screening and diagnosis are routinely offered in antenatal care, and are considered important for managing risk pregnancies and allowing women to make informed choices about the termination of pregnancies affected by DS . A range of policies for prenatal diagnosis have been developed in different countries worldwide, including Romania. Since the introduction of screening policies , most European countries have reported a significant reduction in the rate of liveborns infants with Down syndrome.

In Romania, noninvasive measurements of ultrasound parameters combined with maternal serum markers during the first or second trimester of pregnancy have allowed extension of screening to mothers of all ages, with the use of different software programs for aneuploidy risk calculation. The biochemical markers used in the first trimester of pregnancy (8–13 weeks) are PAPP-A and beta-hCG, and those used in the second trimester of pregnancy (15–18 weeks) are beta-hCG, AFP, and uE3. Different software programs use different techniques for risk calculation using the requested biochemical markers, and present the results in MoM. There are technical standards and recommendations for prenatal DS screening, which include first-trimester biochemistry and/or ultrasound measurements for the first and second trimesters, and can help healthcare professionals and laboratories determine which techniques and screening protocols to choose and how to interpret the results . It is expected that the integration of cffDNA data in the screening algorithm will increase the accuracy of aneuploidy detection, once the cffDNA testing costs becomes affordable or paid by National Insurance. However, even then, more accurate risk assessment will be needed to ensure judicious use of this expensive test. The main limitation of the present study was the small number of cases analyzed. The strengths of this study include the discussion of screening analysis using multiple software programs in a general population of pregnant women, and comparison with the reported cases in other centers.

Round 2

Reviewer 2 Report

The authors have nicely addressed most of my comments